

# SplintR ligation-triggered *in-situ* rolling circle amplification on magnetic bead for accurate detection of circulating microRNAs

Sha Yang[1,2,*], Lijia Yuan[3,*], Xing Luo[1], Ting Xiao[1], Xiaoqing Sun[2], Liu Feng[1], Jiezhong Deng[4] and Mei Zhan[5]

[1] Department of Clinical Laboratory Medicine, Southwest Hospital, The Third Military Medical University, Shapingba District, Chongqing, China

[2] Shigatse Branch, Xinqiao Hospital, The Third Military Medical University, Shigatse, Xizang Province, China

[3] Emergency Department, Southwest Hospital, The Third Military Medical University, Shapingba District, Chongqing, China

[4] Department of Orthopedics, Southwest Hospital, The Third Military Medical University, Shapingba District, Chongqing, China

[5] Nan'an District People's Hospital of Chongqing, Nan'an District, Chongqing, China

[*] These authors contributed equally to this work.

Corresponding authors
Jiezhong Deng,
13350335437@163.com
Mei Zhan, 40443717@qq.com

## ABSTRACT

The circulating microRNAs (miRNAs), endogenous noncoding RNAs, post-transcriptionally participate in multiple processes during cell growth and development. Moreover, dysregulation of miRNAs expression is intricately associated with cancer. Currently, challenges of high homology, sequence similarity, and low abundance encountered in the detection of target miRNAs in complex samples need to be addressed. Biosensors established for miRNAs detection suffer from limitations in terms of sensitivity, specificity and high cost. Herein, a miRNA detection method based on *in-situ* RCA on magnetic bead catalyzed by SplintR ligase was proposed to achieve high sensitivity and high specificity. The following steps are included: (1) formation of P1-P2-miRNA double-stranded complex under catalyzation of SplintR ligase, and the release of P1-P2 single strand under denaturation; (2) enrichment of P1-P2 single chain by streptavidin-modified magnetic beads (SM-MB); (3) *in situ* RCA on surface of magnetic beads; (4) fluorescence detection. After optimization of experimental conditions, miRNA-155 detection with improved sensitivity and specificity was achieved. The detection limit was low to 36.39 fM, and one-base mismatch discrimination was demonstrated. Also, the clinical practicability for circulating miRNA-155 detection was preliminarily validated in human serum samples.

## INTRODUCTION

Circulating microRNA (miRNA) is a class of endogenous non-coding single-stranded RNA with a length of 19 ~ 23 nucleotides (*Lu & Rothenberg, 2018*; *Diener, Keller & Meese, 2022*). MiRNAs could modulate gene expression at the post-transcriptional level

---

and play a variety of regulatory roles in the process of cell growth and development (*Ferragut Cardoso et al., 2021*; *Hill & Tran, 2022*). Research has demonstrated that miRNAs play a crucial role in regulating normal physiological processes *in vivo*, including cell proliferation, differentiation, development, and apoptosis. Furthermore, they are involved in the pathogenesis of various conditions such as tumorigenesis, cardiovascular diseases, and neurological disorders (*Bernardo et al., 2015*; *Hill & Tran, 2021*). Qualitative and quantitative analysis of miRNA is a prerequisite for studying their structure and function (*Mishra, Yadav & Rani, 2016*). The detection of mature miRNAs poses challenges due to their small size, inherent instability and degradation, high sequence homology among family members, as well as their low abundance (*Muniategui et al., 2012*; *Saliminejad et al., 2019*; *He et al., 2022*; *Alexiou et al., 2021*). In addition, the presence of miRNA precursor residues during the preparation of mature miRNA samples introduces interference that compromises the accuracy of the assay (*Chen et al., 2005*).

Currently, a lot of miRNA detection methods have been established, such as the northern blotting, oligonucleotide microarray, and quantitative real-time polymerase chain reaction (qRT-PCR) (*Koscianska et al., 2011*; *Nersisyan et al., 2020*). The previously discussed methods are limited by temporal constraints, high costs associated with reagents and instruments, and inadequate specificity and sensitivity. Consequently, it is essential to develop a rapid, cost-effective method with high sensitivity and specificity for the detection of circulating miRNA.

Nucleic acid detection methods based on DNA ligase or RNA ligase are sequence-specific techniques that primarily relies on the hybridization of two independent oligonucleotide probes to adjacent sites of the target sequence (*Jia et al., 2021*). These methods are frequently utilized in the identification of pathogens and markers of genetic diseases owing to their specificity (*Gansauge et al., 2017*; *Paes Dias et al., 2021*). The SplintR ligase is DNA ligase that specifically catalyze the ligation of adjacent single-stranded DNA segments that fixed by complementary RNA, which makes it highly suitable for detection of miRNAs (*Jin et al., 2016*; *Qin et al., 2023*; *Asa, Ravi Kumara & Seo, 2022*). The point worth noting that binding efficacy of SplintR is more than 100 times higher compared to T4 ligase. Only the 4–6 bp overlap between a DNA probe and miRNA was required for efficient ligation by SplintR Ligase. This property provides more flexibility in designing miRNA-specific ligation probes compared to methods that use reverse transcriptase for cDNA synthesis of miRNA. Rolling circle amplification (RCA) is an isothermal, enzymatic process that catalyzed by certain DNA polymerases during which long single-stranded DNAs are synthesized on a short circular single-stranded DNA template by using a single DNA primer (*Mohsen & Kool, 2016*). RCA has gained great attention and is explored as a critical technique for ultrasensitive detection of RNA, DNA, and protein in diagnostic genomics and proteomics (*Zhang et al., 2006*; *Nilsson et al., 2006*).

The enrichment process is pivotal in the detection of microRNAs (miRNAs) due to their significant homology and sequence similarity. Magnetic microbeads demonstrate exceptional magnetic responsiveness and biocompatibility, facilitating the efficient *in vitro* separation and enrichment of target molecules. When conjugated with antibodies or aptamers, these microbeads can accurately identify and swiftly enrich targets from

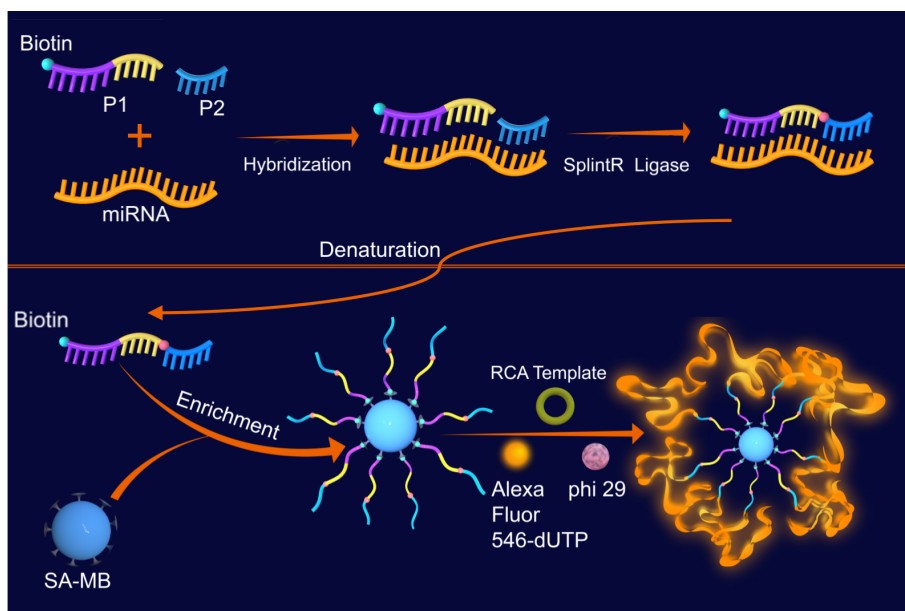

**Figure 1    Schematic graph of the SplintR ligase-based RCA for detection of miRNAs.**

complex biological matrices. Visualization is accomplished through fluorescent labeling techniques, which effectively reduce non-specific adsorption and background interference. Consequently, the application of magnetic beads provides critical technical support for clinical diagnostics and biological analyses.

Taken the limitations of current miRNAs detection methods into account, we developed a miRNA detection method based on SplintR ligase (Fig. 1). Our method achieved single-base mismatch identification of miRNA by combining the SplintR ligase with *in-situ* RCA on magnetic bead, which enhanced the specificity, sensitivity, and detection speed while reducing false positives. The experimental procedures were as follows: (1) formation of P1-P2-miRNA double-stranded complex under catalyzation of SplintR ligase, and the release of P1-P2 single strand under denaturation; (2) enrichment of P1-P2 single chain by streptavidin-modified magnetic beads (SM-MB); (3) *in situ* RCA on surface of magnetic beads; (4) fluorescence detection.

## RESULTS

### SplintR ligase catalyzes the ligation of P1-P2 single stand DNA that target miRNA-155

We performed the non-denatured polyacrylamide gel electrophoresis to determine the interaction between P1-P2 with miR-155. The DNA fragments in lanes 1, 2, and 3 represent P1, P2, and miRNA-155 respectively. The DNAs in lane 4, lane 5, and lane 7 are the P1 and P2, indicating that P1 and P2 cannot be connected without the target miRNA-155 and the ligase (SplintR or T4). This is because in the absence of the target miRNA-155, the two P1 and P2 cannot be linked even in the presence of SplintR ligase or T4 ligase (lane 5

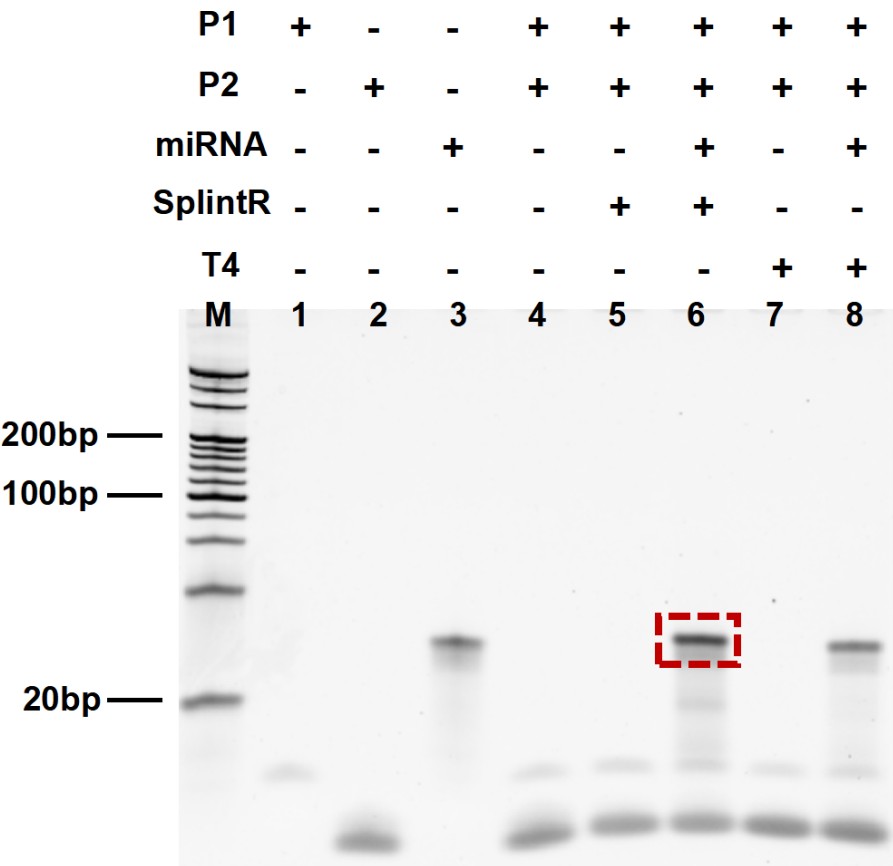

**Figure 2 Results from denatured polyacrylamide gel electrophoresis.** Lane 1: P1, Lane 2: P2, Lane 3: miRNA, Lane 4: P1+P2, Lane 5: P1+P2 with SplintR, Lane 6: P1+P2+miRNA with SplintR, Lane 7: P1+P2 with T4, Lane 8: P1+P2+miRNA with T4.

and lane 7). With the presence of target miRNA-155, P1 and P2 are connected by SplintR ligase or T4 ligase to form a connection product (lane 6 and lane 8). More DNA fragments were generated by SplintR ligase action (lane 6) than by T4 ligase action (lane 8), which proved that SplintR ligase was more efficient and faster than traditional T4 ligase during the same time. These results shown in Fig. 2 demonstrate the feasibility of SplintR ligase in recognizing and binding P1-P2 on miRNA-155.

## TEM characterization of magnetic beads

Next, we used transmission electron microscopy to characterize the reaction products under different conditions, and it can be clearly seen that more reaction products appeared on the surface of the magnetic beads amplified by RCA (Fig. 3C) compared to the other two groups (Figs. 3A, 3B), which demonstrated that the present reaction system can be amplified efficiently and with high sensitivity by RCA in the presence of the target miRNA-155.

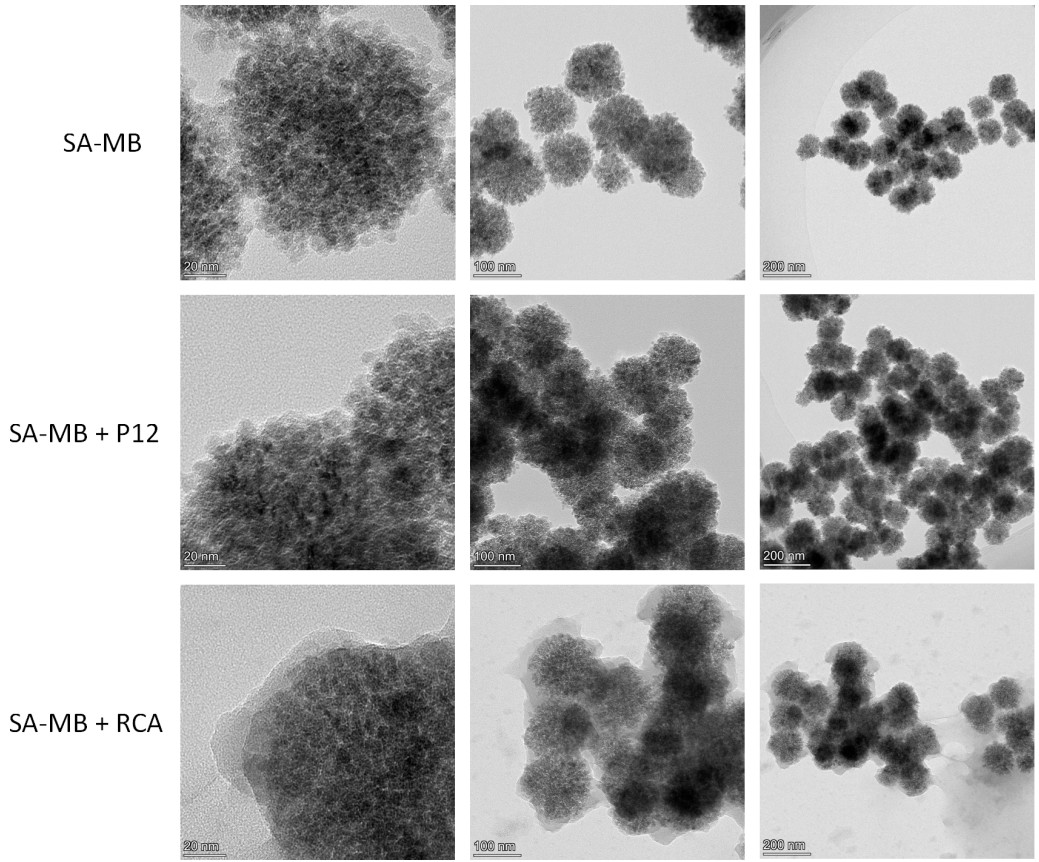

**Figure 3** **TEM characterization of magnetic beads.** (A) Streptavidin-modified magnetic beads (SA-MB). (B) SA-MB enriched with P1-P2 single chain. (C) SA-MB with RCA products.

## Optimization of experimental conditions

We next optimized the experimental conditions to enhance the specificity and sensitivity of the reaction, including the SplintR ligase reaction time, RCA reaction time, and the diameter of streptavidin magnetic bead. The fluorescence intensity measured after adding miRNA-155 was set as F, and the background fluorescence intensity measured with no target miRNA was shown as F0. Then (F-F0)/F0 value was calculated as the index for optimization of experimental conditions. As shown in Fig. 4A, the (F-F0)/F0 value increased with the increase of reaction time and reached a plateau after 60 min. Therefore, 60 min was selected as the SplintR ligase reaction time. Results from Fig. 4B showed that the (F-F0)/F0 value from RCA reached a plateau after 60 min, suggesting the 60 min as optimal RCA reaction time. Moreover, the (F-F0)/F0 value was elevated with the increased diameter of magnetic beads since the size of magnetic beads would interfere with the fluorescence signal (Fig. 4C). Hence, we selected 200 nm streptavidin-modified magnetic beads for subsequent experiments.

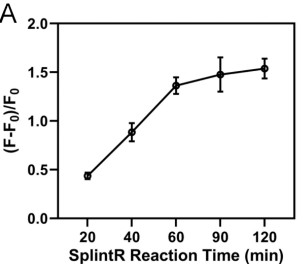
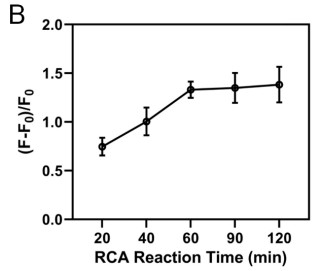
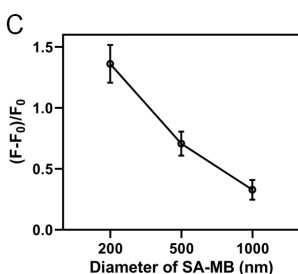

**Figure 4** **Optimization of experimental conditions.** (A) SplintR ligase reaction time. (B) RCA reaction time. (C) The diameter of streptavidin magnetic bead.

## Detection performance

Different concentrations of miRNA-155 were added into reaction system to determine the corresponding fluorescence intensity. As the target concentration increased, the fluorescence intensity (Fig. 5A) and the peak fluorescence intensity (570 nm) (Fig. 5B) both increased. When the concentration of miRNA-155 ranged from 100 fM to one nM, the fluorescence signal response presented a linear relationship (Fig. 5C). We then calculated the linear fitting equation as $F = 1.956 LgC + 1.792$, $R^2 = 0.9991$, and the detection limit is calculated as 36.39 fM based on the $3\delta$/slope. Compared to recent detection methods, our approach exhibits a broad detection range and a low limit of detection (Table 1). The relative standard deviation (RSD) analyses demonstrate that our method exhibits high reproducibility when compared to RT-qPCR (Table 2).

To determine the specificity of this reaction system, we designed single-base mutant miRNA-155 (SMT), double-base mutant miRNA-155 (DMT), three-base mutant miRNA-155 (TMT) and miRNA-122 and miRNA-21 for comparison. The same concentration of these miRNAs (one nM) was added for reaction, and the fluorescence intensity was measured. As shown in Fig. 6, the fluorescence intensity corresponding to the miRNA-155 is significantly higher than that of other corresponding fluorescence intensities, which can be notably distinguished even compared with single base mismatch ($P < 0.01$). These results showed that the detection method of current study has good specificity and can identify single base mismatch. The RSD analyses demonstrate that our method exhibits high reproducibility when compared to RT-qPCR (Table 3).

Moreover, our detection method exhibits a lower cost compared to that of single gene (SG) techniques and TaqMan probe-based assays (Table 4).

## Recovery rate test

We collected serum samples from healthy individuals and diluted them 10-fold to verify the practical utility of this system by detecting five different concentrations of miRNA-155 in the diluted serum samples. The recoveries of this method were 98.31 ~104.27% with RSDs of 9.02 ~13.71% (Table 5). These results indicate that the present system has high accuracy and confidence in detecting complex biological samples.

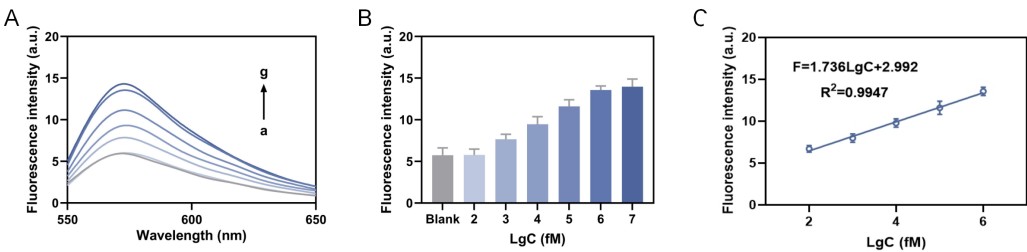

**Figure 5 Detection of sensitivity.** (A) The fluorescence intensity curve. (B) The fluorescence intensity at 570 nm. (C) The linear fitting equation, where the C in LgC is the target concentration.

**Table 1 The detection range and limit of detection (LOD) of detection methods.**

| Method | Detection range | Orders of magnitude | Lod | Ref |
|---|---|---|---|---|
| SERS-Fluorescence | 0.2 to 2 nM | 2 | 11.8 pM | *Wang et al. (2023)* |
| Fluorescence | 0 to 100 nM | 3 | 10 pM | *Xue et al. (2021)* |
| Fluorescence | 0.05 to 2 nM | 3 | 21 pM | *Xing et al. (2022)* |
| Fluorescence | 0.8 to 100 nM | 4 | 0.72 nM | *Deng et al. (2021)* |
| Fluorescence | 1 pM to 1 nM | 4 | 0.77 pM | *Yu et al. (2022)* |
| Fluorescence | 0 to 50 nM | 2 | 1.499 nM | *Zhou et al. (2020)* |
| Fluorescence | 0.5 to 10 $\mu$M | 3 | 2.11 nM | *Gong et al. (2021)* |
| RT-qPCR | 1 pM to 1 nM | 5 | 1 pM | *Chen et al. (2011)* |
| Northern Blot | 1 pM to 1 nM | 2 | 1 pM | *Yang, Zhang & Zhang (2022)* |
| Fluorescence | 100 fM to 1 nM | 5 | 36.39 fM | This work |

**Notes.**
Lod, limit of detection; SERS, surface-enhanced Raman scattering.

**Table 2 Relative standard deviation (RSD) of Fig. 5C.**

| Concentration (LgC/fM) | Relative standard deviation (RSD, %) |
|---|---|
| 2 | 12.07 |
| 3 | 7.99 |
| 4 | 9.79 |
| 5 | 6.84 |
| 6 | 3.58 |

## Clinical application of the proposed biosensor

Afterwards, 20 serum samples were collected from healthy individuals and liver cancer patients respectively. The total miRNAs were collected by kit method (QIAGEN miRNeasy Micro Kit (50), catalog number 217084), miRNA-155 was quantitatively detected by this method and RT-PCR respectively, RT-PCR was carried out with an ABI 7500 real-time PCR instrument (ABI) by protocol. As shown in Fig. 7, the results of demonstrated satisfied
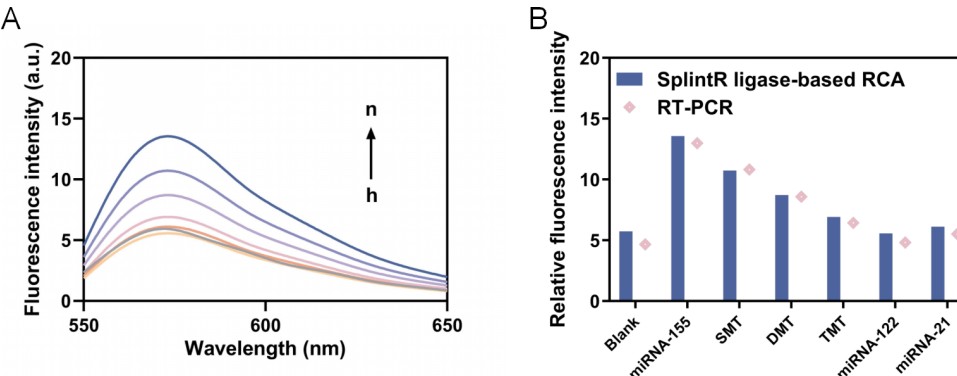

**Figure 6  Detection of specificity.** (A) The fluorescence intensity curve (h to n represents blank, miRNA-122, miRNA-21, TMT, DMT, SMT and miRNA-155, respectively). (B) The fluorescence intensity at 570 nm and relative miRNA expression.

**Table 3  Relative standard deviation (RSD) of Fig. 6B.**

| Samples | Relative standard deviation (RSD, %) |
|---|---|
| Blank | 15.21 |
| miRNA-155 | 3.58 |
| SMT | 5.25 |
| DMT | 4.99 |
| TMT | 3.16 |
| miRNA-122 | 9.11 |
| miRNA-21 | 11.72 |

**Table 4  The detection cost of different methods.**

| Assay | Detection cost ($) | | | |
|---|---|---|---|---|
| | Isolation of RNA | Primers/Probes | Master Mix | Total |
| TaqMan probes | 4 | 20 | 6 | 30 |
| Single-gene (SG) | 4 | 15 | 6 | 25 |
| SplintR ligase | 4 | 1 | / | 5 |

concordance with the gold standard RT-PCR ($R^2 = 0.9291$), thereby further validating the clinical applicability of this approach.

## DISCUSSION AND CONCLUSION

The present study presents a novel miRNA detection approach that synergistically harnesses the strengths of SplintR ligase and RCA. Specifically, SplintR ligase facilitates the recognition of P1 and P2 regions within specific miRNAs, leading to the formation of P1-P2 single-stranded DNA, which subsequently entered the RCA system for fluorescence measurement.
**Table 5** Analytical results for miRNA-155 in diluted human serum samples ($n = 3$).

| Samples | Added | Found | Recovery (%) | RSD (%) |
|---------|-------|-------|--------------|---------|
| Sample 1 | 100 fM | 101.48 fM | 101.48 | 12.91 |
| Sample 2 | 1 pM | 1.01 pM | 101.00 | 13.71 |
| Sample 3 | 10 pM | 9.83 pM | 98.31 | 9.53 |
| Sample 4 | 100 pM | 99.61 pM | 99.61 | 9.02 |
| Sample 5 | 1 nM | 1.04 nM | 104.27 | 10.43 |

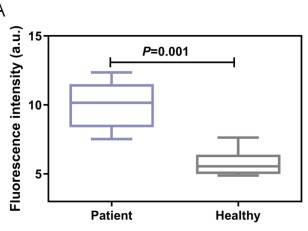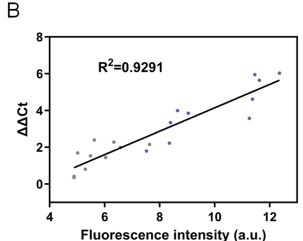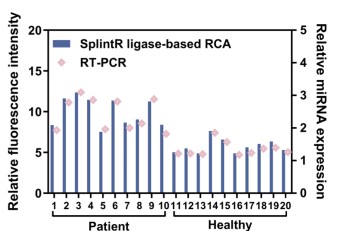

**Figure 7** (A) The fluorescence intensity at 570 nm. (B) Linear correlation analysis between the proposed method and RT-PCR. *** $P < 0.001$. (C) The fluorescence intensity and relative miRNA expression of patient and healthy control.

The single oligonucleotide DNA strands P1 and P2 were designed according to the sequence of miRNA-155. The P1 and P2 can complement the target miRNA sequence completely, and the 3′ OH of P1 chain is close to the 5′ P of P2 chain. SplintR ligase catalyzes the connection of P1 and P2 to form P1-P2-miRNA double-stranded complex. By heating denaturation, the double-stranded complex is released into P1-P2 single strand DNA and miRNA, which avoid the nonspecific amplification during the following RCA. The streptavidin on SA-MB specifically binds to the biotin in P2, hence facilitates the enrichment of P1-P2 single chains. Next, the addition of RCA template and Alexa fluor 546-conjugated dNTPs initiated the RCA cycle on surface of magnetic beads, which is the second process of this detection process. The fluorescence intensity will increase proportionally with the abundance of RCA products in the presence of fluorescent dye conjugated dNTP.

The SplintR ligase has recently emerged as a valuable tool for genomic analysis. A previous study designed a two-step method for miRNA detection utilizing SplintR Ligase-based qPCR, during which miRNA specific probes were ligated to target miRNA followed by qPCR detection. In qPCR reactions, amplified DNA can be detected by binding to a dye, such as SYBR Green that binds to all dsDNA, or by using specific DNA probe that contains a quencher and fluorophore (*Jin et al., 2016*). SplintR ligase was also applied in a detection method named as SPC that based on the amplification of tertiary signals using splintR ligase ligation, PCR amplification and CRISPR/Cas12a cleavage (*Zhou et al., 2023*). Here, we combined SplintR ligase with RCA for detection of miRNAs to utilize their advantages to elevate the specificity and sensitivity. Our work for the first time combined the SplintR

ligase and magnetic beads-based RCA, realized the high sensitivity and specificity for miRNA detection. The P1-P2 single strand elevated the specificity and SA-MB-based RCA elevated the sensitivity.

The proposed biosensor offers the following advantages: (1) Traditional RCA method is prone to generating a significant number of non-specific amplification products in the presence of complex templates or low primer content, thereby reducing detection efficiency. The proposed strategy involving SplintR DNA ligase for substrate recognition, along with subsequent magnetic bead enrichment in this method, ensures the abundant RCA primer templates output. This dual recognition and amplification mechanism guarantees both high sensitivity and specificity. (2) The entire detection process can be conducted at ambient temperature (approximately 25 °C), without requiring intricate operations or expensive analyzers. Moreover, the DNA structure of the miRNA sensor is straightforward and can be easily adapted for detecting other types of RNA with minor modifications. The integration of portable fluorescence detection devices enhances the potential of POCT due to its relatively straightforward procedural steps. (3) The dye was conjugated to the magnetic beads together with RCA, rather than being re-conjugated after the formation of RCA product, thereby significantly enhancing the binding efficiency.

Our proposed sensor still requires further improvement. For instance, the utilization of non-one-step detection may lead to partial loss of nucleic acid samples and an increased likelihood of contamination during operation, particularly in point-of-care testing (POCT) conditions. Moreover, the inevitable interference between magnetic materials and fluorescence in challenge that needs to be addressed for achieving compatibility between the two.

The sensor based on magnetic beads and RCA has high specificity and sensitivity compared to traditional detection methods, *e.g.*, alpha-fetoprotein (AFP) lacks specificity for the detection of hepatocellular carcinoma because acute and chronic hepatitis and other malignant tumours also cause it to be elevated. Therefore, this method may be applied to more detection areas, such as lncRNA, circRNA, mRNA, etc. In conclusion, our study presents a straightforward, rapid, and cost-effective method for the detection of miRNAs in clinical samples. The assay demonstrated a lower limit of detection (36.39 fM), excellent specificity (ability to detect single-base mismatches), satisfactory recovery rate and good consistency with PT-PCR. It is noteworthy that the utilization of a dual-amplification mechanism in this study yielded a detection limit comparable to that of SERS and electrochemical biosensors, thereby overcoming the limitation of insufficient detection limits observed in fluorescent biosensors (*Sun et al., 2022*; *Ma et al., 2024*; *Yu et al., 2023*). At the same time, we believe that the simplicity of its structural design and sequence design enhance the universality of the biosensor, enabling it to detect various types of RNA. In addition, the utilization of magnetic bead targeting also enables precise on-site detection studies, thereby presenting extensive prospects for application in clinical and rapid detection.
**Table 6   Sequences of primers and miRNA.**

| Name | Sequences (5′–3′) |
| --- | --- |
| P1 | Biotin-ACCCCTATCAC |
| P2 | P-GATTAGCATTAA |
| miRNA-155 | UUAAUGCUAAUCGUGAUAGGGGU |
| RCA Template | TTAATAACCACAAGCCCAAAACTCGAGTCTGATAATA CCCAAATAGATTA (TTAAT is complementary to P2) |
| SMT | UUAAUGCUAAUCGUGAUCGGGGU |
| DMT | UUAAUGCUAAUCGUGGUCGGGGU |
| TMT | UUAAUGCUAAUCGUGGUCGGGAU |
| miRNA-155-F | AAGCGCCTTTAATGCTAATCGT |
| miRNA-155-R | CAGTGCAGGGTCCGAGGT |
| miRNA-155-RT | GTCGTATCCAGTGCAGGGTCCGAGGTATTCGCACT GGATACGACAACCCC |
| U6-F | AGAGAAGATTAGCATGGCCCCTG |
| U6-R | ATCCAGTGCAGGGTCCGAGG |
| U6-RT | GTCGTATCCAGTGCAGGGTCCGAGGTAT TCGCACTGGATACGACAAAATA |

**Notes.**
    "P" in sequence P2 stands for 5′-phosphorylation.

# MATERIALS AND METHODS

## SplintR ligase catalyzed formation of P1-P2 single strand

All oligonucleotides (Table 6), purified *via* high-performance liquid chromatography (HPLC), were obtained from Sangon Biotech Co., Ltd. (Shanghai, China). A mixture of one μL P1 (10 μM), one μL P2 (10 μM), one μL miRNA-155 (10 μM), one μL SplintR ligase (25 U/L, NEB, USA) and two μL 10× SplintR ligase reaction buffer (50 mM Tris–HCl, 10 mM $MgCl_2$, one mM ATP, 10 mM DTT, pH 7.5; NEB) were added into 14 μL $ddH_2O$. The reaction was performed under the conditions of 16 °C for 60 min and 95 °C for 5 min. The products were then divided by a 12% non-denatured polyacrylamide gel at 120 V for 45 min. Similarly, reaction by T4 DNA ligase as the control was conducted following the same steps.

## Enrichment of P1-P2 single strand by magnetic beads

Streptavidin (SA) on the magnetic bead specifically interacted with biotin on the P1 chain, which enriched the P1-P2 single chain. In brief, a total of 10 μL streptavidin-modified magnetic beads (SA-MB; 10 mg/mL, 200 nm in diameter) were washed in 30 μL magnetic bead buffer (20 mM Tris–HCl, 1.0 M NaCl, 0.02% Triton® X-100; pH 7.8) for three times and then re-suspended in 30 μL magnetic bead buffer. Then reaction solution of P1-P2 single strand was added to the magnetic beads and were slowly shaken for 10 min, followed by magnetic separation of magnetic beads, and washed with $ddH_2O$ for three times.

## *In situ* RCA

The purified magnetic beads were added into a reaction buffer that composed of one μL RCA Template (10 μM), 10 μL Alexa fluor 546-dUTP (one mM), one μL dATP (10 mM),

**Table 7** Characteristics of the study participants.

| Characteristics | Age (years) | Gender (male/female) | Fluorescence intensity (a.u.) |
|---|---|---|---|
| Healthy individuals | $62.90 \pm 7.31$ | 7/3 | $5.78 \pm 0.88$ |
| Liver cancer patients | $61.80 \pm 7.83$ | 7/3 | $10.00 \pm 1.76$ |
| $P$ | $>0.05$ | $>0.05$ | $<0.001$ |

one $\mu$L dGTP (10 mM), and one $\mu$L dCTP (10 mM), 2two$\mu$L BSA, one $\mu$L phi29 DNA polymerase (10 U/L, NEB, USA), 2.5 $\mu$L phi29 DNA polymerase reaction buffer (50 mM Tris–HCl, 10 mM MgCl2, 10 mM $(NH_4)_2SO_4$, four mM DTT, pH 7.5, NEB, USA) and 5.5 $\mu$L ddH$_2$O at 30 °C for 60 min and 65 °C for 10 min. The RCA template was designed to be complementary to P2 sequence.

## TEM characterization of magnetic beads

The magnetic beads enriched with P1-P2 single chain were subjected to RCA reaction, after the reaction was completed, the reaction products were mixed well, and 15 uL of the reaction products were taken on the copper mesh and left to stand for 2 min, then the reaction products on the copper mesh were sucked up using filter paper, and then the copper mesh was stained with 2% hydrogen peroxide acetate staining solution at room temperature for 1 min. When the copper mesh was seen to be adsorbable, pure water could be used to repeatedly add drops to the surface, and then the filter paper was used to suck up the reaction products, and then the photographs were taken under observation. When the adsorption is visible on the copper mesh, the surface can be repeatedly dribbled with pure water and finally dried with filter paper and photographed under observation.

## Detection of fluorescence

After the *in-situ* RCA reaction, MB with RCA products was magnetically separated. And then 75 $\mu$L ddH$_2$O was added and mixed. The fluorescence intensity was recorded with excitation wavelength at 500 nm and emission wavelength at 550~650 nm.

## RT-qPCR

A 2X SG Fast qPCR master mix was used according to the manufacturer's protocol, and the thermal cycling conditions were as follows: an initial denaturation at 95 °C for 3 min, followed by 40 cycles of denaturation at 95 °C for 3 s and annealing/extension at 60 °C for 30 s. A U6 small nuclear RNA was utilized as an internal reference for normalization of gene expression data in all samples analyzed.

## Statistical analysis

All data are expressed as mean $\pm \beta$ standard deviation based on data from three independent experiments. SPSS software (version 20.0) was employed for statistical analyses. Statistical significances were determined using Student's $t$-tests for comparing two groups, and analysis of variance (ANOVA) for comparing multiple groups.

## Clinical application

The serum samples from healthy individuals and liver cancer patients were collected from the Third Medical Military University Southwest Hospital with approval by the ethics committee of the hospital (approval number: KY2020146). Our study utilized residual samples obtained from routine clinical examinations of patients, which were confirmed by the ethics committee to not require informed consent. The characteristics of the study participants are shown in Table 7.

### Funding

This work was supported by the National Key Research and Development Program of China (No. 2022YFC2603800) and the National Natural Science Foundation of China (No. 82030066). The funders had no role in study design, data collection and analysis, decision to publish, or preparation of the manuscript.

### Grant Disclosures

The following grant information was disclosed by the authors:
National Key Research and Development Program of China: No. 2022YFC2603800.
National Natural Science Foundation of China: No. 82030066.

### Competing Interests

The authors declare there are no competing interests.

### Author Contributions

- Sha Yang conceived and designed the experiments, prepared figures and/or tables, authored or reviewed drafts of the article, and approved the final draft.
- Lijia Yuan performed the experiments, prepared figures and/or tables, authored or reviewed drafts of the article, and approved the final draft.
- Xing Luo performed the experiments, prepared figures and/or tables, authored or reviewed drafts of the article, and approved the final draft.
- Ting Xiao analyzed the data, prepared figures and/or tables, and approved the final draft.
- Xiaoqing Sun analyzed the data, prepared figures and/or tables, and approved the final draft.
- Liu Feng conceived and designed the experiments, prepared figures and/or tables, authored or reviewed drafts of the article, and approved the final draft.
- Jiezhong Deng conceived and designed the experiments, prepared figures and/or tables, authored or reviewed drafts of the article, and approved the final draft.
- Mei Zhan conceived and designed the experiments, prepared figures and/or tables, authored or reviewed drafts of the article, and approved the final draft.

### Human Ethics

The following information was supplied relating to ethical approvals (i.e., approving body and any reference numbers):

The serum samples from healthy individuals and liver cancer patients were collected from the Third Medical Military University Southwest Hospital with approval by the ethics committee of the hospital (approval number: KY2020146).

## Data Availability

The raw measurements are available in the Supplementary Files.

## Supplemental Information

Supplemental information for this article can be found online at http://dx.doi.org/10.7717/peerj.19082#supplemental-information.

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
