# Peer review of "SplintR ligation-triggered in-situ rolling circle amplification on magnetic bead for accurate detection of circulating microRNAs"

_PeerJ, doi:10.7717/peerj.19082_

## Round 0.1 · original submission · Major Revisions

· Academic Editor

Major Revisions

Please address issues pointed by both reviewers and amend manuscript accordingly.

Reviewer 1 ·

Basic reporting

The authors describe an in situ-RCA on the surface of magnetic bead method to detect the expression of miR-155 using a SplintR Ligase to catalyze the reactions. The text could be improved, as some English errors were found, suggestion it should be revised by an English native speaker. The literature review could be improved by citing similar methods that were already described using SplintR Ligase to detect miRNAs. Importantly, it should be clear which are the innovations presented by this current method compared to similar ones. Figures and tables are clear, although their citation in the text has some mistakes and I suggest additional information should be added. Despite this, the article has relevant results and is suitable for publication if the authors address some issues that are described in this report.

Experimental design

The experimental design is adequate, but the methods lack information, as described in the next section.

Validity of the findings

The authors mention the high cost of the currently used miRNA detection methods and suggest that the newly proposed method would be less expensive. I suggest that they clarify this by adding the actual and detailed costs of their method in comparison to the use of TaqMan miRNA detection probes. Importantly, the authors do not describe the RT-qPCR method used as the gold standard : which primers and probes were used? Which master mix? What were the cycling conditions? The method to obtain RNA is not detailed (in line 150 : the total miRNA was collected by kit method - what is the kit? Which type of RNAs it recovers? ).
In line 187: The text describes the results shown in Figure 6, but refers to Figure 5. In this same paragraph, the authors described that they added 1 ng of each miRNA (155, 155 SMT, DMT, TMT and miR-122 and miR-21) to the sample, in order analyze specificity. I would suggest that the results of RT-PCR were shown in parallel to the proposed method, with specific probes to miR-155.
Figure 7 should contain the complete results of the RT-qPCR and of the proposed method to allow comparison. Also, the legend of Figure 7 lacks information to understand the figure.

Reviewer 2 ·

Basic reporting

The manuscript is generally clear, but some sentences are lengthy and could be broken down for improved readability. For example, the sentence describing the experimental procedures in the abstract is dense and could be divided for better flow.

The article follows a standard scientific structure with clear sections (Abstract, Introduction, Methods, Results, Discussion, and Conclusion). This is appropriate for the intended audience.

Suggestions for Improvement: Perform a language edit for minor grammatical errors and sentence structure.

Experimental design

The manuscript addresses an original research question, which is highly relevant to molecular biology and diagnostic technologies. The combination of SplintR ligation and RCA for circulating microRNA detection fills a notable gap in existing methods, particularly in terms of sensitivity, specificity, and cost-effectiveness.

While the manuscript highlights that SplintR ligase is more efficient than T4 ligase, a detailed comparison of the ligation rate or error rate is not provided. Can the authors elaborate on the specific advantages of SplintR ligase over other ligases, including T4 ligase and dsDNA ligase?

Suggestions for Improvement:
1)Justify the clinical sample size or perform a power analysis to ensure statistical robustness.
2)Provide additional details on fluorescence detection settings and statistical analyses to enhance reproducibility.
3) Include a comparison of this method with existing miRNA detection technologies in the discussion to contextualize the findings.

Validity of the findings

The study presents a novel combination of SplintR ligation and RCA for miRNA detection, which is robustly validated through experimental results. While direct replication of similar methods is not performed in this study, the rationale for developing this method and its benefits to the field are clearly explained, particularly the need for a sensitive, specific, and cost-effective alternative to conventional detection techniques like qPCR.

However, the inclusion of a side-by-side comparison with other established miRNA detection methods (e.g., qPCR, northern blotting, or biosensors) would provide additional confidence in the broader applicability and superiority of this approach.

Furthermore, the optimization of experimental conditions (e.g., reaction times, bead sizes) was performed specifically for miRNA-155. It is unclear whether these optimized conditions are universally applicable to other targets or whether significant re-optimization would be required for different miRNAs.

The conclusions are clearly stated and well-supported by the data. The authors summarize the advantages of the method (e.g., high specificity, sensitivity, cost-effectiveness, and ease of use) and acknowledge the potential for its application in detecting other RNA targets, such as lncRNAs or circRNAs.

Suggestions for Improvement:
1) Specify the statistical tests and software used for analyzing specificity and recovery rate data.
2) Discuss the adequacy of the clinical sample size for statistical significance and its potential limitations.
3) Provide more detailed information on any efforts to mitigate potential sources of variability (e.g., fluorescence interference by magnetic beads).
4) Expand the discussion on how this study contributes to advancing the field of nucleic acid diagnostics and its potential applications beyond miRNA-155 detection.

Additional comments

There are several points that need to be clarified by the authors:

1)Does the RCA-based detection method using magnetic beads offer significant improvements in speed or signal intensity compared to RCA without magnetic beads? It would be helpful for the authors to include a direct comparison experiment or provide data showing the benefits of using magnetic beads, such as improved enrichment or reduced background signal.

2)How reliable is this technique when implemented in different laboratory settings or by different operators? Have the authors conducted inter-laboratory validation or tested the method's reproducibility across multiple batches of reagents and equipment? Including such data or discussing potential variability would provide greater confidence in the method's robustness.

---

## Round 0.2 · Minor Revisions

· Academic Editor

Minor Revisions

Please address the remaining concerns of the reviewer and amend your manuscript accordingly.

Reviewer 1 ·

Basic reporting

The authors described an in situ-RCA on the surface of magnetic bead method to detect the expression of miR-155 using a SplintR Ligase to catalyze the reactions. The text could was improved now, as compared with the first version. The literature review was also improved and the innovations of the current method were added to the text. Figures and tables were corrected.

Experimental design

The experimental design is adequate and the authors made the corrections as suggested.

Validity of the findings

The authors added a table (4) with the comparison of the costs for each methodology, but I still suggest that the costs are described in details. For example, cost of isolation of RNA per reaction, TaqMan probes cost, master mix cost, if reactions are in duplicate, etc.

The sequence of primers and probes was added as well as a more detailed description of RNA isolation, as suggested.
I have suggested in the first revision that the results of RT-PCR were shown in parallel to the proposed method, with specific probes to miR-155 and the authors added a graph with fluorescence intensity for each RT-PCR, including miR-155. Despite this, I would still recommend that the real time PCR amplification curves were demonstrated, not only for miR-155 but also for the endogenous controls, describing in detail which one(s) were used (and add the sequences of those controls to Table 2).

Figure 7 is adequate now.

Annotated reviews are not available for download in order to protect the identity of reviewers who chose to remain anonymous.

Reviewer 2 ·

Basic reporting

After thoroughly assessing the revised version, I find that the authors have adequately addressed the comments and suggestions provided during the initial review. The improvements made to the manuscript are substantial and align well with the feedback previously given.

Experimental design

no comment

Validity of the findings

no comment

---

## Round 0.3 · accepted · Accept

· Academic Editor

Accept

All remaining concerns of the reviewer were addressed, and the revised manuscript is acceptable now.